# Interpolating Video-LLMs: Toward Longer-sequence LMMs in a Training-free Manner

## Abstract

Advances in Large Language Models (LLMs) have inspired various strategies for integrating video modalities. A key approach is Video-LLMs, which incorporate an optimizable interface linking sophisticated video encoders to LLMs. However, due to computation and data limitations, existing Video-LLMs are typically pre-trained to process only short videos, limiting their broader application for understanding longer video content. Additionally, fine-tuning Video-LLMs to handle longer videos is cost-prohibitive. Consequently, it is essential to explore the interpolation of Video-LLMs under a completely training-free setting. In this paper, we first identify the primary challenges in interpolating Video-LLMs: ❶ the video encoder and modality alignment projector are fixed, preventing the integration of additional frames into Video-LLMs, and ❷ the LLM backbone is limited in its content length capabilities, which complicates the processing of an increased number of video tokens. To address these challenges, we propose an **INT**er**P**olation method for Video-LLMs (`INTP`-Video-LLMs). We introduce a video token rearrangement technique that circumvents limitations imposed by the fixed video encoder and alignment projector. Furthermore, we introduce a training-free LLM context window extension method to enable Video-LLMs to understand a correspondingly increased number of visual tokens. We analyze the deployment costs of `INTP`-Video-LLM, and find its efficiency bottleneck is on its KV cache cost. Accordingly, we introduce a training-free KV-cache compression mechanism that reduces memory overhead during inference. `INTP`-VideoLLM not only supports the processing of longer video sequences but also optimizes memory usage during inference—all achieved without the need for additional training. In practice, whereas pre-trained Video-LLaVA [Lin et al., 2024] models are configured to process just 8 frames, `INTP` allows these models to comprehend 32 frames.

## 1 Introduction

Large Language Models (LLMs) [OpenAI, 2023, Touvron et al., 2023a;b], have shown tremendous capabilities in question answering and reasoning. Building on this foundation, Vision LLMs extend these abilities to include images, employing a vision encoder and an LLM to generate text responses given an image and a related question [Liu et al., 2023a, Zhang et al., 2024]. Recent advancements aim to extend this capability from image understanding to video understanding [Lin et al., 2024, Zhang et al., 2023a, Kim et al., 2024]. Video-LLMs combine video data with language models by integrating learnable interfaces that capture both spatial and temporal information in video. Typically, these interfaces use a projection network [Li et al., 2023b, Maaz et al., 2023, Li et al., 2023a] to transform video content into video tokens that can be interpreted by LLMs, thereby bridging the gap between video information and text processing capabilities of LLMs (see Fig.1).

However, existing Video-LLMs [Li et al., 2023b, Lin et al., 2024, Maaz et al., 2023, Li et al., 2023a, Kim et al., 2024] struggle to process long videos. In particular, Video-LLMs often lack the temporal resolution necessary to precisely model temporal events, limiting its application to long video [Huang et al., 2024]. For instance, Video-LLaMA [Li et al., 2023b] and Video-LLaVA [Lin et al., 2024] only sample 8 frames uniformly across an entire video, which is often too short for detailed temporal analysis [Huang et al., 2024]. Simply feeding more frames into these models does not resolve the issue, as their architectures cannot effectively process a larger number of frames. Consequently, there is a pressing need for Video-LLMs that can understand larger numbers of video frames.

Figure 1: **(Left) Video-LLMs** consist of three main components: a video encoder, an alignment projector layer, and a fine-tuned LLM backbone. The process begins with the video encoder transforming video frames into a series of visual tokens. A projector then maps these tokens, aligning visual features with text features. The resulting aligned features, along with text prompts, are subsequently fed into the LLM backbone for visual understanding. **(Right) INTP**, a training-free Video-LLM interpolation technique, addresses the existing constraints of Video-LLMs. We employ a video token **rearrangement** that bypasses the limitations set by the fixed video encoder and alignment projector. Additionally, we implement a training-free LLM context window **interpolation** method to allow Video-LLMs to process an increased number of visual tokens effectively.

To reach this goal, one could consider training Video-LLMs from scratch to handle more extensive video frames. Unfortunately, this approach can be impractical due to prohibitive training costs and data accessibility issues. On the one hand, Video-LLMs require increasingly large datasets, making data acquisition more challenging, especially considering copyright restrictions on video-language paired data. In some cases, the training data are not publicly available. Sometimes, only the model weights are released while the data is not open. On the other hand, the computation cost of training a Video-LLM is expensive. For example, even the relatively less costly Video-LLaVA [Lin et al., 2024] requires thousands of hours on Nvidia A100 GPUs. Therefore, retraining a Video-LLM to process more frames is not feasible in many scenarios.

In this paper, we aim to significantly increase the number of frames that existing Video-LLMs can use to understand videos in a training-free manner. We first delineate the primary challenges associated with interpolating Video-LLMs: ❶ the video encoder and modality alignment projector are fixed, which prevents the integration of additional frames, and ❷ the content length capacity of the LLM backbone is limited, hindering the processing of an increased number of visual tokens.

To overcome these obstacles, we propose an `interpolation` method for Video-LLMs, dubbed `INTP`. For the ❶ challenge, we propose a video token rearrangement technique that bypasses the restrictions imposed by the fixed video encoder and alignment projector. This approach allows utilizing the pre-trained video encoder and alignment projector to generate an increased number of video tokens for LLM reasoning while maintaining the video tokens' temporal consistency. For the ❷ challenge, derived from the positional embedding mechanism in Video-LLMs (i.e., Rotary Position Embedding, RoPE [Su et al., 2024]), we develop a training-free Video-LLM context window extension method. This design ensures that the interpolated Video-LLM can handle any number of video frames.

Upon developing the `INTP`-Video-LLM, we examine its deployment constraints and identify that the handling of an increased number of video tokens incurs additional memory usage. This finding echos the memory bottleneck in long-sequence LLM deployment [Yuan et al., 2023; 2024]. To address this issue and enhance the deployment efficiency of `INTP`-Video-LLM, we introduce a training-free KV-cache compression technique that reduces memory overhead during inference. Consequently, `INTP`-Video-LLM not only facilitates the processing of longer video sequences but also optimizes memory usage during inference, all without necessitating further training.

## 2 RELATED WORK

### 2.1 VIDEO LANGUAGE MODELS

Prior to the emergence of LLMs, Yang et al. [2022] introduced FrozenBiLM, which combined a frozen vision encoder with a bidirectional language model for efficient video processing. In the LLM era,

video-language models continue to evolve. For instance, Video-ChatGPT enhances video instruction tuning using high-quality instructional data [Maaz et al., 2023]. Li et al. [2023a] propose VideoChat, which utilizes cross-attention mechanisms to integrate video tokens with user queries and dialogue context. Li et al. [2024] propose a follow-up work, VideoChat2, which advances this integration with a multi-stage bootstrapping method, focusing on modality alignment and further instruction tuning. Video-LLaVA [Lin et al., 2024] introduces a pre-aligned encoder that supports both image and video modalities, enabling shared projections and joint training across tasks. However, handling long videos is very challenging due to the high computational complexity and memory demands of representing extensive video content with video tokens. To address these issues, various advanced temporal modeling techniques have been deployed. Chat-UniVi [Jin et al., 2023] proposes a unified model that dynamically merges spatial and temporal tokens using k-NN techniques to streamline processing. LLaMA-VID [Li et al., 2023b] uses a dual token system to effectively compress video tokens. Despite these explorations, all previous methods rely on some form of training. In contrast, our work is the first to explore the extension of the Video-LLM context window in a *training-free* manner, offering an approach to handling long video sequences without the computational expense and data requirements of traditional training methods.

## 2.2 LONG-CONTEXT LLMs

LLMs are generally pre-trained with a specified context length, with models like LLaMA and LLaMA2 employing 2k and 4k tokens respectively [Touvron et al., 2023a;b]. Training LLMs from scratch to handle extended context is often prohibitively expensive for many researchers. Thus, recent innovations have focused on extending the context length through fine-tuning methods. Recent advancements focus on modifying the position embedding (PE), particularly RoPE [Su et al., 2024], employed in LLMs such as LLaMA and Mistral [Jiang et al., 2023]. One of the main strategies is embedding scaling [Chen et al., 2023, Liu et al., 2023b, Peng et al., 2023], which modifies position embeddings to accommodate longer sequences, ensuring they remain within the pre-training scope and preventing feature extrapolation. For instance, Chen et al. [2023] condense position indices to maintain alignment with the pre-training range, effectively expanding LLaMA's context to 16,000 tokens with minimal fine-tuning, requiring only 1,000 steps. In a different approach, Liu et al. [2023b], Roziere et al. [2023] adjust the rotary base of RoPE, known as "NTK-aware" scaling. To the best of our knowledge, our work is the first exploration to extend the context window of Video-LLMs in a training-free manner.

## 2.3 POST-TRAINING COMPRESSION FOR KV CACHE

Recently, the compression of Key-Value (KV) caches has garnered significant attention due to the prohibitively high memory consumption associated with generating long contextual sequences in LLMs. Current methods can be briefly categorized into three types: Quantization-aware, eviction-based, and attention-based. Quantization-aware compression reduces KV cache size by substituting the original KV cache with lower-precision approximations. KVQuant [Hooper et al., 2024], KIVI [Liu et al., 2024] and WKVQuant [Yue et al., 2024] are pioneering studies in KV cache quantization, revealing that keys and values should be quantized along different dimensions. Notably, KIVI compresses the KV cache to as few as 2 bits. Yang et al. [2024], Dong et al. [2024] further enhance quantization performance by identifying and selectively preserving more significant keys and values. SKVQ [Duanmu et al., 2024] advances upon KIVI using a sliding window technique, wherein quantization parameters are determined on a window-wise basis. ZipCache [He et al., 2024] establishes a robust baseline for KV cache quantization, achieving compression ratios as high as 5x with negligible performance degradation. In this work, we analyze the development bottlenecks of long-context Video-LLMs and identify that the major constraint lies in the LLM's KV cache. To address this computational overhead, we leverage quantization techniques, effectively optimizing the model's memory usage and inference efficiency.

## 3 METHOD: INTERPOLATING VIDEO-LLMs

In this section, we first review the standard implementation of Video Large Language Models (Video-LLMs), with a particular focus on the main components and training pipelines. We highlight the difficulties of obtaining a long-video-LLM from scratch (Sec.3.1). Next, we present a totally

training-free method specifically designed for Video-LLM **INT**er**P**olation, called INTP. There are three advancements in the development and optimization of INTP: (1) A video encoder and modality-alignment projector extension method (Sec.3.2); (2) A video-LLM context window extension method (Sec. 3.3); and (3) An inference bottleneck analysis and a post-training quantization solution (Sec.3.4).

## 3.1 PRELIMINARIES: VIDEO LARGE LANGUAGE MODELS (VIDEO-LLMS)

Video-LLMs are typically composed of three core modules: a video encoder, an alignment projector layer, and a fine-tuned large language model (LLM) backbone. The process begins with a video encoder—often a Vision Transformer (ViT)—which converts video input $\mathbf{X}_v \in \mathbb{R}^{N \times W \times H}$ into a series of visual tokens $\mathbf{Z}_v$, in which $N$ is number of frames, $W$ and $H$ is the pixel width and height of each frame. These tokens are processed by the input projector $\mathbf{\Theta}_{\mathbf{Z}_v \to \mathbf{H}_v}$, which maps the encoded visual features to the text feature space $\mathbf{H}$. The aligned features, along with text prompts $\mathbf{H}_q$, are then input into a LLM backbone, and then the language model can generate the response. The architecture of a typical Video-LLMs is illustrated in Fig. 1 (left).

Video encoders are crucial for compressing the raw video into compact representations. These encoders are often trained from scratch via multimodal alignment pretraining [Yin et al., 2023]. For instance, LanguageBind [Zhu et al., 2023] employs contrastive learning during the pretraining phase to gradually align video modality with language modality. Given that LLMs inherently process text, bridging the modality gap between natural language and video is essential. However, training a Video-LLM from scratch is prohibitively expensive. A more feasible approach involves using a learnable connector between the pre-trained visual encoder and the LLM, along with fine-tuning the LLM to better interpret visual tokens. Although those components can be trained via instruction tuning [Gong et al., 2023] or alignment tuning [Sun et al., 2023], their training remains costly. For example, to train the video encoder, Zhu et al. [2023] collect a large-scale video-and-text dataset, VIDAL-10M. In addition, even without pre-training the video encoder, fine-tuning the Video-LLaVA [Lin et al., 2024] still consumes approximately 200 hours on Nvidia A100 GPUs.

In summary, given the significant data and computational expenses associated with training Video-LLMs, developing these models from scratch to process long videos is challenging. This underscores the importance of *exploring methods to adapt existing Video-LLMs for extended video processing in a training-free manner*.

In practice, the challenges in extending the Video-LLMs can be summarized from two key aspects: ❶ the *fixed nature of the video encoder and modality alignment projector*, and ❷ the *limited content length capacity of the LLM backbone*. First, the video encoder and alignment projector are typically configured during initial training and remain unchanged thereafter. This prevents the system from incorporating additional frames beyond the preset limit, which limits the model's ability to adapt to videos of different lengths or to capture more granular temporal details within prolonged video sequences. Second, the LLM backbone has a predefined limit on the number of tokens it can handle. This constrains the model's capacity to interpret a larger number of visual tokens, effectively capping the amount of visual information that can be processed. As the complexity and length of videos increase, these constraints become significant bottlenecks, hindering the model's ability to fully understand and generate coherent responses based on longer video inputs.

## 3.2 ALTERNATIVE VIDEO TOKEN REARRANGEMENT

To address challenge ❶, we enable the video encoder and alignment projector to map video inputs with an increased number of frames into the text feature space. As shown in Fig. 2, the pre-trained video encoder is configured to process a fixed-length video sequence, $\mathbf{X}_v$, consisting of $N$ frames. The most straightforward approach is repeatedly using the encoder and projector to process $m \cdot N$ frames by sampling and encoding multiple subsets of frames $m$ times. Once we have collected $m$ groups of video tokens, we can concatenate them together to get the whole long video's visual tokens.

However, since these video encoders and projectors are pre-trained to process only $N$ frames collectively, such repetitive usages lead to distorted temporal representations in the video token sequence. The concatenating video tokens may carry the frame's relative information—such as the inconsistency between the tokens of the $N$-th frame and those of the $(N + 1)$-th frame (i.e., tokens of the 1-th frame and those of the $(N + 1)$-th frame end up sharing more similarity). This discrepancy

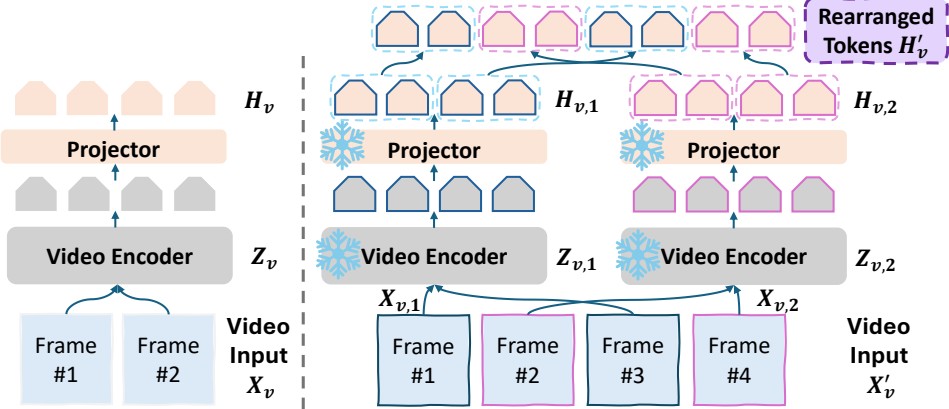

Figure 2: **Alternative Video Token Rearrangement.** (Left) A video input sampled with fewer frames, such as 2 frames $\mathbf{X}_v$, is processed through a video encoder and projector to produce visual tokens $\mathbf{Z}_v$ and transformed features $\mathbf{H}_v$. (Right) Increasing the number of sampled frames, for example to 4 (Frame #1 - #4), results in a richer video input $\mathbf{X}'_v$. By pairing Frames #1 with #3 and #2 with #4, we obtain two subsequences, $\mathbf{X}_{v,1}$ and $\mathbf{X}_{v,2}$, each processed by the same frozen encoder and projector. This results in new sets of tokens ($\mathbf{Z}_{v,1}$, $\mathbf{H}_{v,1}$; $\mathbf{Z}_{v,2}$, $\mathbf{H}_{v,2}$). These tokens are then correspondingly integrated into an extended sequence of features $\mathbf{H}'_v$.

emerges because the transformer-processed tokens are not designed to naturally accommodate the increased spatial and temporal variety, potentially causing misalignments and inconsistencies in the encoded video features [Xiao et al., 2023, Shang et al., 2024a].

To overcome this issue, we propose a video token rearrangement technique that preserves the temporal consistency of the video tokens. The technique allows the generation of an increased number of temporal-consistent video tokens, which enable the model to process extended video sequences without requiring retraining. The module employs a dynamic rearrangement of input frames and their corresponding tokens. Here's how the rearrangement works:

1. For a given video input $\mathbf{X}_v$ with $N$ frames, the video encoder generates an initial set of visual tokens $\mathbf{Z}_v$. These tokens are subsequently mapped by the alignment projector to produce $\mathbf{H}_v$. Both the encoder and the projector remain frozen to maintain consistency in processing.

2. To enhance temporal coverage, the number of frames sampled is increased to $m \cdot N$, and the video is divided into subsequences to capture various temporal segments. For example: The first subsequence might include Frames $\{\#1, \#(m+1), \cdots \#(mN - m + 1)\}$, providing some new but temporally overlapped "view" of these segments. Similarly, the rest $i$-th subsequence might consist of Frames $\{\#i, \#(m+i), \cdots \#(mN - m + i)\}$. These subsequences are processed separately through the same fixed video encoder and projector: The $i$-th subsequence processes to yield visual tokens $\mathbf{Z}_{v,i}$ and corresponding projected tokens $\mathbf{H}_{v,i}$. This approach increases the amount of visual data that can be processed without modifying the architecture or requiring retraining of the encoder and projector.

3. The visual tokens generated from each frame group are correspondingly integrated into a new sequence based on their frame's absolute location, $\mathbf{H}'_v$, as illustrated in Fig. 2.

This newly assembled token sequence can represent a more extended duration of the video than a single processing pass of the original input would allow. In the following subsection, we demonstrate how to adapt the Video-LLM backbone to process the increased number of video tokens.

### 3.3 INTERPOLATING VIDEO-LLM BACKBONE

To address challenge ❷, we introduce a post-training interpolation method for the Video-LLM's backbone, specifically targeting its positional embedding. As discussed in the related work (Sec. 2), Rotary Position Embedding (RoPE) [Su et al., 2024] is a widely-used positional encoding scheme adopted by several prominent LLMs, including Llama, Phi, Mistral, and Gemma. Consequently, RoPE has become the standard method for position embedding in Video-LLMs.

### 3.3.1 RoPE Fundamentals

RoPE integrates positional information into both the query vector $\mathbf{q}$ and key vector $\mathbf{k}$ of transformer models. Specifically, for a sequence of tokens $s_1, s_2, \cdots, s_L$ with embeddings $\mathbf{x}_1, \cdots, \mathbf{x}_L \in \mathbb{R}^D$ ($D$ is the dimension of the embedding), RoPE applies the following transformations:

$$\mathbf{q}_m = f_q(\mathbf{x}_m, m) \in \mathbb{R}^D \quad \text{and} \quad \mathbf{k}_n = f_k(\mathbf{x}_n, n) \in \mathbb{R}^D. \tag{3.1}$$

The transformation functions $f_q$ and $f_k$ incorporate positional information as:

$$f_q(\mathbf{x}_m, m) = e^{im\theta} \mathbf{W}_q \mathbf{x}_m \quad \text{and} \quad f_k(\mathbf{x}_n, n) = e^{in\theta} \mathbf{W}_k \mathbf{x}_n, \tag{3.2}$$

where $\theta_d = b^{-2d/|D|}$ with a base $b = 10000$. These transformations ensure that the relative positions of tokens are reflected in their interactions through the inner product operations, represented by $m - n$ of the tokens as follows: $\langle f_q(\mathbf{x}_m, m), f_k(\mathbf{x}_n, n) \rangle_{\mathbb{R}} =$

$$\text{Re}(\langle f_q(\mathbf{x}_m, m), f_k(\mathbf{x}_n, n) \rangle_{\mathbb{C}}) = \text{Re}(\mathbf{x}_m^* \mathbf{W}_q^* \mathbf{W}_k \mathbf{x}_n e^{i\theta(m-n)}) = g(\mathbf{x}_m, \mathbf{x}_n, m - n) \tag{3.3}$$

where $*$ is the conjugate of a complex number, $\text{Re}(\mathbf{q}, \mathbf{k})$ is the real part of the inner product of $\mathbf{q}$, $\mathbf{k}$, and $g(\cdot)$ is an abstract mapping function. In real coordinates, the RoPE can be expressed as follows:

$$f_{\mathbf{W}}(\mathbf{x}_m, m, \theta_d) = \begin{pmatrix} \cos m\theta_1 & -\sin m\theta_1 & 0 & 0 & \cdots & 0 & 0 \\ \sin m\theta_1 & \cos m\theta_1 & 0 & 0 & \cdots & 0 & 0 \\ 0 & 0 & \cos m\theta_2 & -\sin m\theta_2 & \cdots & 0 & 0 \\ 0 & 0 & \sin m\theta_2 & \cos m\theta_2 & \cdots & 0 & 0 \\ \vdots & \vdots & \vdots & \vdots & \ddots & \vdots & \vdots \\ 0 & 0 & 0 & 0 & \cdots & \cos m\theta_l & -\sin m\theta_l \\ 0 & 0 & 0 & 0 & \cdots & \sin m\theta_l & \cos m\theta_l \end{pmatrix} \mathbf{W}\mathbf{x}_m. \tag{3.4}$$

### 3.3.2 Context Window Scaling

Given the fixed context lengths of pre-trained Video-LLMs, a significant question is how to extend these lengths cost-effectively. Utilizing the inherent flexibility of RoPE, we modify the embedding functions to handle extended sequences without extensive retraining:

$$f'_{\mathbf{W}}(\mathbf{x}_m, m, \theta_d) = f_{\mathbf{W}}\left(\mathbf{x}_m, \frac{mL}{L'}, \theta_d\right), \tag{3.5}$$

where $L' > L$ denotes a new, extended context window. It allows us to adapt the pre-trained positional embeddings to longer contexts, maintaining accuracy.

### 3.3.3 NTK-aware Interpolation

The "NTK-aware" interpolation further refines this adjustment by recalibrating the base of the RoPE function:

$$\theta' = b'^{-2d/|D|} \quad \text{and} \quad b' = b \cdot s^{\frac{|D|}{|D|-2}}, \tag{3.6}$$

where $s = \frac{L'}{L}$ is the scaling ratio. This adjustment ensures that the extended embeddings retain their functionality within the longer context length [Roziere et al., 2023].

By leveraging these techniques, we can extend the context window of Video-LLM's backbone, enhancing their capacity for understanding longer video in a training-free manner.

### 3.4 Efficiency Bottleneck Analysis and a Post-training Solution

Implementing the Alternative Video Token Rearrangement (Sec. 3.2) and the Interpolating Video-LLM Backbone (Sec. 3.3) lead to the development of an interpolated Video-LLM (INTP-Video-LLM). Indeed, this enhances long-video understanding capabilities without incurring additional training costs. This advancement, however, raises a crucial question: *does INTP-Video-LLM introduce any additional costs during the model inference phase?*

Table 1: Computation Cost Analysis. The development device is A100 GPU, and time estimated by the roofline model represents the theoretical performance that the hardware can achieve.

| Method | Frame Number | LLM Backbone | Quantization | OPs (TB) | Decode Time (ms) | Total Memory (GB) | Storing KV (GB) |
|---|---|---|---|---|---|---|---|
| Video-LLaVA | 8 | Vicuna-7B | FP16 | 14.2 | 18.6 | 14.0 | 1.1 |
| INTP-Video-LLaVA | 16 | Vicuna-7B | FP16 | 15.3 | 20.1 | 15.1 | 2.1 |
| INTP-Video-LLaVA | 16 | Vicuna-7B | INT2 | 15.3 | 17.6 | 13.2 | 0.3 |
| INTP-Video-LLaVA | 32 | Vicuna-7B | FP16 | 17.4 | 22.9 | 17.2 | 4.3 |
| INTP-Video-LLaVA | 32 | Vicuna-7B | INT2 | 17.4 | 22.9 | 13.5 | 0.5 |
| INTP-Video-LLaVA | 64 | Vicuna-7B | FP16 | 21.8 | 28.6 | 21.5 | 8.6 |
| INTP-Video-LLaVA | 64 | Vicuna-7B | INT2 | 21.8 | 18.8 | 14.0 | 1.1 |
| INTP-Video-LLaVA | 128 | Vicuna-7B | FP16 | 30.4 | 39.9 | 30.1 | 17.2 |
| INTP-Video-LLaVA | 128 | Vicuna-7B | INT2 | 30.4 | 20.4 | 15.1 | 2.1 |

### 3.4.1 INFERENCE COST ANALYSIS

To quantify the inference costs, we employ a modified roofline-based LLM-Viewer analysis, initially developed in [Yuan et al., 2024]. Consider a typical scenario in which each frame is represented by 256 visual tokens. In standard Video-LLaVA [Lin et al., 2024] configurations, 8 frames represent a video, totaling 2048 visual tokens. With the implementation of INTP, more frames can be processed, escalating the number of visual tokens significantly. This increase is thoroughly analyzed in Tab. 1, which details the computational cost implications for the LLM backbone's decoding process, especially focusing on the model latency and memory cost.

The analysis indicates that the primary additional cost during the inference phase for INTP-Video-LLaVA is due to an increased demand for KV-cache storage, a requirement driven by the larger volume of visual tokens being processed. This finding aligns with studies in long-context LLMs [Fu, 2024, Ashkboos et al., 2024], which highlight that the decoding phase of LLMs is predominantly memory-bounded. This underscores the critical need for effective KV-cache management strategies to ensure that the enhanced capabilities of INTP-Video-LLaVA can be deployed efficiently.

To address this issue, we introduce a KV-cache compression technique aimed at optimizing KV-cache storage efficiency. **Post-training quantization (PTQ)** is utilized to convert full-precision tensor inputs into quantized tensors, focusing primarily on selecting optimal quantization parameters. The process requires only two parameters: the scaling factor $S$ and the zero point $Z$, which are crucial for the quantization process [Shang et al., 2024b, Yuan et al., 2024]. Formally, once appropriate values for $S$ and $Z$ are determined, a full-precision key (or value) tensor can be quantized as follows:

$$\mathbf{K}_Q = S(\text{clamp}(\lfloor \frac{\mathbf{K}_F}{S} \rceil - Z, p_{min}, p_{max}) + Z), \tag{3.7}$$

where $[p_{min}, p_{max}]$ is the quantization range determined by bit-width, for 2 bit integer, and the bins are $\{-2, -1, 0, 1\}$. However, direct application of this method to KV quantization is not straightforward because KV tensors require the input to be accessed. To address this, we employ a calibration dataset (a significantly smaller set of input samples compared to the training dataset) to collect the necessary tensors. Once the full-precision KV tensors at the $k$-th layer, $\mathbf{K}_F^k$ and $\mathbf{V}_F^k$, are obtained, they can be quantized using the same method to produce $\mathbf{K}_Q^k$ and $\mathbf{V}_Q^k$ based on Eqn. 3.7.

By implementing this quantization technique, INTP-Video-LLM can manage the increased volume of visual tokens during inference, ensuring that the model's enhanced capabilities are aligned with practical deployment constraints.

## 4 EXPERIMENTS

We detail the experimental setup and model configurations in Sec. 4.1, followed by an analysis of the quantitative and qualitative performance of our approach in Sec. 4.2 and Sec. 4.4. Finally, we ablate the effectiveness of each component in our model in Sec. 4.3.

Table 2: A comparison of different LVMs on video reasoning benchmarks. Like Video-LLaVA [Lin et al., 2024], ChatGPT-Assistant is employed to evaluate the following performance. The version of ChatGPT is "gpt-3.5-turbo".

| Methods | LLM size | Num Frames | MSVD-QA Accuracy | MSVD-QA Score | MSRVT-QA Accuracy | MSRVT-QA Score | ActivityNet-QA Accuracy | ActivityNet-QA Score |
|---------|----------|-----------|------------------|---------------|-------------------|----------------|-------------------------|----------------------|
| FrozenBiLM [Yang et al., 2022] | 1B | 8 | 32.2 | - | 16.8 | - | 24.7 | - |
| VideoChat [Li et al., 2023a] | 7B | 8 | 56.3 | 2.8 | 45.0 | 2.5 | - | 2.2 |
| LLaMA-Adapter [Zhang et al., 2023b] | 7B | - | 54.9 | 3.1 | 43.8 | 2.7 | 34.2 | 2.7 |
| Video-LLaMA [Zhang et al., 2023a] | 7B | 8 | 51.6 | 2.5 | 29.6 | 1.8 | 12.4 | 1.1 |
| Video-ChatGPT [Maaz et al., 2023] | 7B | 8 | 64.9 | 3.3 | 49.3 | 2.8 | 35.2 | 2.7 |
| Video-LLaVA [Lin et al., 2024] | 7B | 8 | 70.7 | 3.9 | 59.2 | 3.5 | 45.3 | 3.3 |
| Video-LLaVA+INTP | 7B | 32 | **72.0** +1.3 | **4.0** +0.1 | **61.4** +2.2 | **3.5** +0.0 | **48.9** +3.6 | **3.5** +0.2 |

Table 3: Performance results on multiple-choice question benchmarks.

| Methods | LLM size | Ego-Schema | NExT-QA Cas. | NExT-QA Tem. | NExT-QA Des. | NExT-QA Avg. |
|---------|----------|-----------|--------------|--------------|--------------|--------------|
| FrozenBiLM [Yang et al., 2022] | 1B | 26.9 | - | - | - | - |
| InternVideo [Wang et al., 2024] | 1.3B | 32.1 | 48.0 | 43.4 | 65.1 | 59.1 |
| Sevilla [Yu et al., 2024] | 7B | - | 61.3 | 61.5 | 75.6 | 63.6 |
| Video-ChatGPT [Maaz et al., 2023] | 7B | - | 61.9 | 57.4 | 69.9 | 61.7 |
| Video-LLaVA [Lin et al., 2024] | 7B | 37.0 | 61.2 | 54.2 | 71.1 | 60.5 |
| Video-LLaVA+INTP | 7B | **38.6** +1.6 | **61.9** +0.7 | **58.6** +4.4 | **72.2** +1.1 | **62.7** +2.2 |

## 4.1 EXPERIMENTAL SETUP

### 4.1.1 MODEL SETTINGS

We use the Video-LLaVA [Lin et al., 2024] framework to explore Video-LLM interpolation.[1] We employ Vicuna-7B v1.5 as the LLM backbone, mirroring the architecture used in Video-LLaVA. The visual encoders are adapted from LanguageBind, and the text tokenization is handled by the LLaMA tokenizer, which includes approximately 32,000 classes. The projection layers consist of 2 fully connected layers, facilitating the integration of visual and textual data.

### 4.1.2 DATA AND TRAINING DETAILS

Consistent with our focus on developing a training-free method, our approach requires no traditional data training processes. This aspect underscores the innovative nature of our interpolation method, which leverages existing model architectures and tools without the need for retraining or additional data, simplifying the implementation and reducing the computational overhead typically associated with training new models.

### 4.1.3 DATASETS AND EVALUATION METRIC

We assess INTP across a variety of zero-shot video question-answer benchmarks, which we classify as either open-ended or multiple-choice, depending on the type of question posed [Kim et al., 2024]. Our evaluation for open-ended VQA includes MSVD-QA [Xu et al., 2017], MSRVTT-QA [Xu et al., 2017], and ActivityNet-QA [Yu et al., 2019] benchmarks. We utilize GPT-assisted assessments as outlined in Video-ChatGPT [Maaz et al., 2023], which provide a thorough evaluation of the model's response accuracy and correctness. For multiple-choice VQA tasks, we evaluate the model's performance on NExT-QA [Xiao et al., 2021] and EgoSchema [Mangalam et al., 2024]. We determine accuracy by the model's ability to correctly select the appropriate answer from the provided options.

---

[1] We develop our method based on the Video-LLaVA codebase. The evaluation of our approach is conducted using the IG-VLM [Kim et al., 2024] codebase.

Table 4: Ablation Studies on using different numbers of frames of INTP.

| Methods | Number of Frames | MSVD-QA | | MSRVT-QA | | ActivityNet-QA | |
|---|---|---|---|---|---|---|---|
| | | Accuracy | Score | Accuracy | Score | Accuracy | Score |
| Video-LLaVA [Lin et al., 2024] | 8 | 70.7 | 3.9 | 59.2 | 3.5 | 45.3 | 3.3 |
| Video-LLaVA+INTP | 8 | 69.5 | 3.9 | 58.2 | 3.5 | 55.3 | 3.3 |
| Video-LLaVA+INTP | 16 | **72.1** +1.4 | **4.0** +0.1 | 61.0 | 3.5 | 46.9 | 3.3 |
| Video-LLaVA+INTP | 32 | 72.0 | **4.0** +0.1 | **61.4** +2.2 | **3.5** +0.0 | **48.9** +3.6 | **3.5** +0.2 |
| Video-LLaVA+INTP | 64 | 67.5 -3.2 | 3.8 -0.2 | 55.2 -4.0 | 3.3 -0.2 | 41.5 -3.8 | 3.1 -0.2 |

## 4.2 QUANTITATIVE RESULTS

The experimental results for three open-ended VQA benchmarks are detailed in Tab. 2, and the results for two multiple-choice VQA benchmarks are presented in Tab. 3. These results demonstrate that our training-free INTP enhances the performance of existing Video-LLMs. Notably, INTP acts as a plug-and-play enhancement, boosting Video-LLM capabilities without incurring additional costs. This compatibility underscores INTP's potential as a universally applicable upgrade for current Video-LLM frameworks.

## 4.3 ABLATION STUDIES

We consider Alternative Video Token Rearrangement and Interpolating Video-LLM Backbone as one unit for realizing a long-video-LLM. Therefore, in our ablation study, we assess the impact of varying the number of frames processed by INTP-Video-LLaVA. The results are presented in Tab. 4. This analysis leads to two main conclusions: **(i, Performance Improvement with Increased Frames)** A general improvement in performance as the number of frames increases, validating the effectiveness of the INTP. By feeding more frames into INTP-Video-LLMs are able to comprehend more extensive video content, thereby enhancing their performance. Note that INTP is realized on the pre-trained Video-LLMs in a training-free manner. **(ii, Performance Plateau at Higher Frame Number)** However, a performance plateau is observed when the number of frames is scaled to 64. This suggests a limitation in the NTK-based LLM backbone extension method, which appears to struggle with significantly expanding the LLM content window capacity beyond a certain point. This finding highlights a potential area for future research and optimization to further enhance the capacity of Video-LLMs to handle long video sequences effectively.

## 4.4 QUALITATIVE RESULTS

Finally, we visualize the question-and-answer samples within the ActivityNet dataset to assess the qualitative performance of our approach in Fig. 3. In the first example, while Video-LLaVA incorrectly describes a person "riding a bike on a skateboard" and misidentifies the hair length, INTP-Video-LLaVA accurately describes the scene as a person "wearing orange clothes and jumping on stilts on a trampoline" and correctly states that the person does not have long hair. This shows INTP-Video-LLaVA's superior ability to process and interpret a larger number of video frames, leading to more accurate scene description and detail recognition. In the second example, Video-LLaVA hallucinates the presence of a second person playing a guitar, which is not shown in the video frames. In contrast, INTP-Video-LLaVA correctly identifies that there are two people in the video, accurately describes the main activity (playing drums), and even provides specific details about the types of drums being played (congas and tabla). Additionally, INTP-Video-LLaVA correctly identifies the gender of the person playing the drums, demonstrating its improved ability to process and understand visual information across a longer sequence of frames. These examples highlight how INTP-Video-LLaVA's capacity to handle more video frames leads to more accurate, detailed, and coherent descriptions of video content, significantly reducing errors and hallucinations compared to the standard Video-LLaVA model.

## 5 CONCLUSION

In conclusion, we presented INTP, an innovative interpolation method for Video-LLMs that effectively addresses the limitations of current models in processing extended video sequences. By developing an alternative video token rearrangement technique and a training-free LLM context

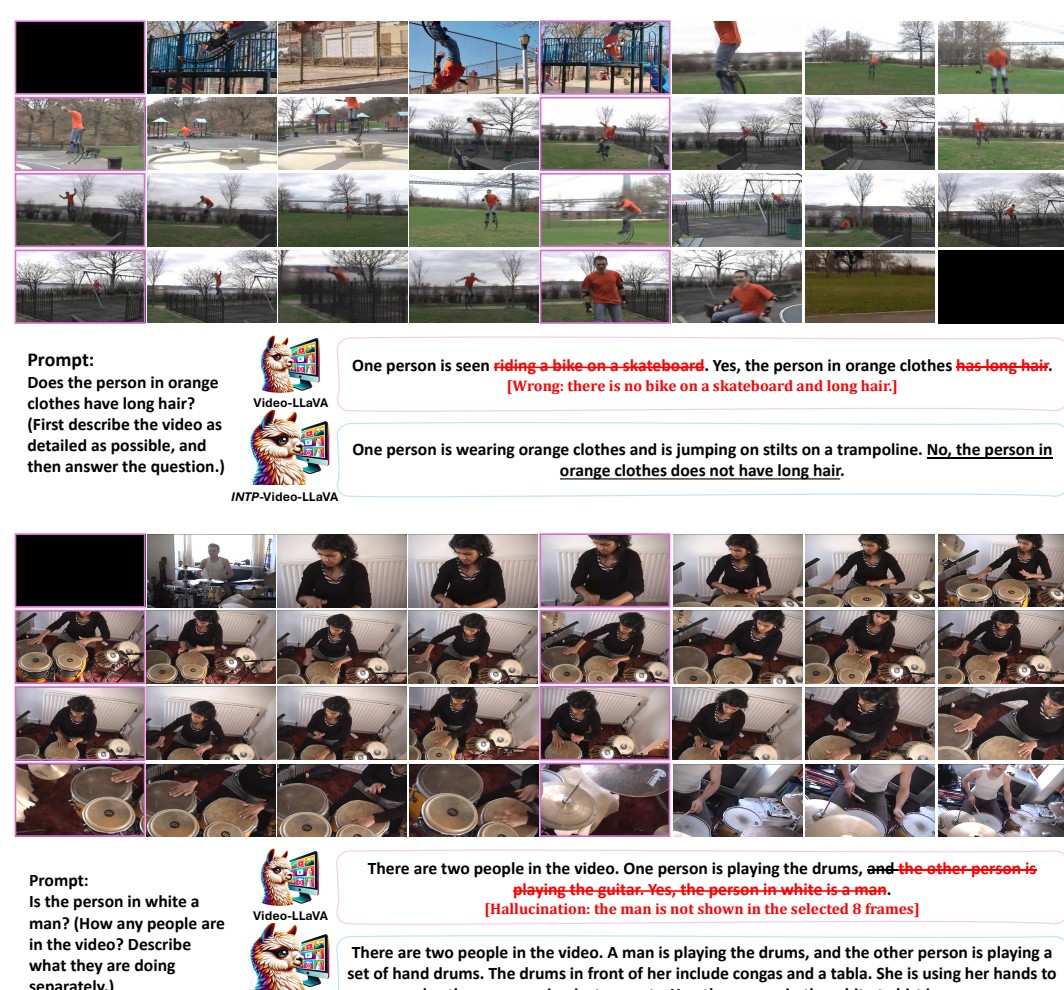

Figure 3: Video content and questions from ActivityNet [Yu et al., 2019]. The standard Video-LLaVA model exhibits limitations in accurately answering the questions, primarily due to its inability to process a sufficient number of frames (only the pink labeled frames are fed into Video-LLaVA). This constraint significantly hinders its effectiveness in managing complex video question-answering tasks. With our proposed INTP, the enhanced Video-LLM can process extended sequences of video frames. This not only addresses the frame limitations but also substantially enhances the model's understanding in complex video question-answering scenarios.

window extension, INTP enables Video-LLMs to handle significantly more visual data (32 frames) without the need for additional training or substantial computational resources. Furthermore, the introduction of a KV-cache compression module optimizes memory usage during inference, enhancing deployment efficiency. These advancements not only extend the practical utility of Video-LLMs but also demonstrate the potential of training-free approaches in advancing Video-LLMs.

**Societal Impacts.** Our approach can democratize the access to advanced video processing technologies by reducing computational demands. Despite these advancements, it is important to note that INTP does not address potential security concerns related to the misuse of Video-LLMs. The ease of accessibility and enhanced capabilities could potentially be exploited by malicious actors to generate or manipulate video content in harmful ways. Thus, while INTP expands the practical applications of Video-LLMs and reduces barriers to their use, it also necessitates careful consideration of the ethical implications and security measures associated with their deployment.

**Reproducibility Statement.** Our method is implemented using the Video-LLaVA codebase, with evaluations conducted via the IG-VLM framework. As a training-free approach, our interpolated model can directly use existing Video-LLM weights, enhancing reproducibility. We will publicly release our code upon the paper's acceptance to facilitate further research in this area.

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
