# OpenReview forum: "Interpolating Video-LLMs:  Toward Longer-sequence LMMs in a Training-free Manner"
_ICLR.cc/2025/Conference — Submitted to ICLR 2025_

### Official Review · Reviewer_qZSz · 2024-10-28

**Soundness:** 3
**Presentation:** 2
**Contribution:** 2
**Rating:** 5
**Confidence:** 3

**Summary:**

This paper presents a interpolation method for extending Video-LLMs to process longer video sequences under training-free setting, called INTP-Video-LLMs. The approach leverages a video token rearrangement technique and a training-free LLM context window extension method to bypass the limitations of existing Video-LLMs, which typically only process short video clips. Furthermore, a training-free key-value (KV) cache compression mechanism is introduced to optimize memory usage during inference. The proposed INTP-Video-LLMs can comprehend longer video sequences (up to 32 frames) without additional training, and experimental results indicate that this approach provides a significant improvement in both video processing capabilities and inference efficiency.

**Strengths:**

- The motivation for the proposed INTP-Video-LLM is clear and fundamental. The Introduction is well-crafted, making it easy to grasp the paper's concept.
- The proposed methods allow existing Video-LLMs to process longer video (i.e. more video frames) **in a training-free manner**, effectively addressing computational constraints.
- The paper comprehensively considers and analyzes whether the proposed **Video Tokens Rearrangement** and **Interpolating Video-LLM Backbone** will lead to additional computational overhead.
- The **memory optimization** via KV-cache compression ensures that extended video sequences can be processed with minimal memory overhead, making the approach feasible for practical deployment.

**Weaknesses:**

- The paper points out the existing limitations in the temporal consistency of encoders and projectors, but the description of the video token rearrangement method is unclear. It is not explained why rearrangement would help maintain consistency, and a more thorough explanation is needed.
- Near Figure 2, at line 234, it describes “we obtain two subsequences, $X_{v,1}$ and $X_{v,1}$,” where $X_{v,1}$ appears twice. Is this a typographical error?
- In Section 3.4.1, the authors analyze the inference cost and propose 2-bit quantization of the KV cache to reduce storage overhead during inference. However, the potential performance degradation due to quantization is not discussed in detail. It would be beneficial to include an analysis of the trade-offs between reduced storage and potential accuracy loss.
- In the ablation study (Section 4.3), the authors compare the performance of using different numbers of frames in Video QA tasks. However, it is unclear what the individual contributions of each module are to the final performance. A detailed breakdown of the impact of each module would provide more insight into their effectiveness.

**Questions:**

Q1. The authors perform 2-bit quantization of the KV cache to reduce storage overhead during the inference process of the video LLM. However, does quantizing the stored KV result in performance degradation?

Q2. Could you provide a more detailed description of the video tokens rearrangement process (e.g., in the form of pseudocode) as well as an analysis of the effectiveness of this method?

---

> ### Author Response · Authors · 2024-11-24
> **Response to Reviewer qZSz (I): Video Token Rearrangement Process**
>
> ## W1 Detailed Description of the Video Token Rearrangement Process
>
> Our goal is to enable the pre-trained video encoder and alignment projector, which are configured to process a fixed number of frames $N$, to handle longer video sequences without retraining. The challenge arises because directly concatenating tokens from multiple passes of the encoder can lead to temporal inconsistencies, as the encoder's internal representations are not designed to naturally accommodate increased temporal variety.
>
> To address this, we propose a video token rearrangement technique that preserves temporal consistency. The key idea is to interleave frames across multiple subsequences and then rearrange the resulting tokens to reflect the correct temporal order. Here's a step-by-step description:
>
> 1. Sampling Frames: Given a video sequence with $m\times N$ frames, we aim to process it using the fixed-capacity encoder. We divide the total frames into $m$ subsequences, each containing $N$ frames.
>
> 2. Constructing Subsequences: Instead of dividing the video into consecutive chunks, we sample frames in an interleaved manner to capture temporal dynamics across the entire video. For the $i$-th subsequence $(i=0,1,\cdots, m-1)$, we select frames at positions:
> $FrameIndices = \{i+jm| j=0,1,\cdots, N-1\}$. This means each subsequence contains frames spaced $m$ frames apart, ensuring that all parts of the video are represented.
>
> 3. Processing Subsequences: Each subsequence $\mathbf{X}_{v,i}$ is processed independently through the frozen video encoder and
> projector:
>
>      $Z_{v,i} = VideoEncoder(X_{v,i})$, $H_{v,i} = Projector(Z_{v,i})$. This yields $m$ sets of visual tokens $\mathbf{H}_{v,i}$, each corresponding to a subsequence.
>
> 4. Reassembling Tokens: We rearrange the tokens from all subsequences to reconstruct the full temporal sequence:
>       - For each frame position $k = 0, 1, \ldots, mN-1$, we identify the subsequence $i = k \bmod m$ and the position within that subsequence $j = \lfloor \frac{k}{m} \rfloor$.
>       - The token for frame $k$ is then $\mathbf{H}_{v,i}[j]$.
>
>     - This process results in a new token sequence $\mathbf{H}'_v$ that aligns with the original temporal order of the video frames.
>
> ## Q2 Pseudocode Illustration  of the Video Token Rearrangement Process
> To further clarify, we provide pseudocode for the video token rearrangement process:
>
>
> ```python
> # Assume we have a list of frames: frames = [frame_0, frame_1, ..., frame_{mN-1}]
> # Number of frames per subsequence: N
> # Number of subsequences: m
>
> # Step 1: Initialize subsequences
> subsequences = [[] for _ in range(m)]
>
> # Step 2: Distribute frames into subsequences in an interleaved manner
> for idx, frame in enumerate(frames):
>     subseq_idx = idx % m
>     subsequences[subseq_idx].append(frame)
>
> # Now, each subsequence[i] contains frames at positions:
> # subsequence[0]: frames at indices [0, m, 2m, ..., (N-1)*m]
> # subsequence[1]: frames at indices [1, m+1, 2m+1, ..., (N-1)*m+1]
> # ...
> # subsequence[m-1]: frames at indices [m-1, 2m-1, 3m-1, ..., mN-1]
>
> # Step 3: Process each subsequence through the encoder and projector
> tokens_list = []
> for subseq in subsequences:
>     Z_v = video_encoder(subseq)  # Visual tokens from the encoder
>     H_v = projector(Z_v)         # Projected features
>     tokens_list.append(H_v)
>
> # Step 4: Rearrange tokens to reconstruct the full sequence
> H_v_prime = []
> for frame_idx in range(len(frames)):
>     subseq_idx = frame_idx % m
>     token_idx = frame_idx // m
>     H_v_prime.append(tokens_list[subseq_idx][token_idx])
>
> # H_v_prime now contains tokens ordered according to the original frame sequence
> ```
>
> ## Explanation of Why Our Method Leads to Better Temporal Consistency
> **Interleaved Frame Sampling**: By constructing subsequences where frames are sampled every $m$ frame (i.e., frames are spaced $m$ frames apart within each subsequence), we ensure that each subsequence matches the temporal structure the encoder expects.
>
> **Problem with Concatenation**: Simply concatenating tokens from multiple encoder passes can introduce temporal discontinuities because the encoder processes each chunk independently, without awareness of the temporal context across chunks.
>
> **Comprehensive Temporal Coverage**: Interleaving frames allows each subsequence to capture temporal information spread throughout the entire video, rather than just a segment.
>
> **Compatibility with the Encoder's Training, Frozen Encoder and Projector**: Since the encoder and projector are kept frozen, it's
> important to provide inputs that align with their training conditions.

---

> > ### Author Response · Authors · 2024-11-24
> > **Response to Reviewer qZSz (II)**
> >
> > ## W2 Typo Correction
> >
> > Thank you for catching this typographical error. We have corrected the text near Figure 2.
> >
> > ---
> > ## W3&Q1 Clarification on KV Cache Quantization
> > **Implementation Details:** We adopted KIVI [1] for 2-bit KV-cache quantization. This method can achieve consistent accuracy across all Video-LLM benchmarks. The method's success in maintaining performance aligns with findings in general LLM KV cache quantization.
> >
> > We would like to note that KV cache quantization is not our core technical contribution. Rather, it complements our training-free interpolation method by addressing the efficiency bottleneck. This forms part of our complete solution for practical deployment of extended-context Video-LLMs.
> >
> > [1] A Tuning-Free Asymmetric 2bit Quantization for KV Cache, ICML 2024
> >
> > ---
> > ## W4 Clarification on Ablation Studies
> > Thank you for suggesting these important ablation studies. We have conducted comprehensive experiments to analyze each component's contribution, with results shown below:
> > Method	| MSVD-QA	| MSRVT-QA | 	ActivityNet-QA
> > | --- | --- | --- | --- |
> > Video-LLaVA |	70.7	| 59.2 | 45.3
> > Video-LLaVA+INTP  |	72.0 (+1.3) |	61.4 (+2.2)	| 48.9 (+3.6)
> > Video-LLaVA+INTP w/o token rearrangement |	68.2 (-3.8) |	60.2 (-1.2) |	44.5 (-4.4)
> > Video-LLaVA+INTP w/o NTK |	Failed	| Failed |	Failed
> >
> > These results reveal several key insights:
> > Necessity of NTK-aware Interpolation: Without NTK-aware scaling, the model fails to generate meaningful responses. This demonstrates that proper position embedding interpolation is critical for extending the context window while maintaining model coherence.
> > Importance of Token Rearrangement: Removing the token rearrangement module leads to significant performance drops (3.8% on MSVD-QA, 4.4% on ActivityNet-QA), indicating its crucial role in maintaining temporal relationships when processing longer sequences.
> > Synergistic Effect: The full INTP system shows consistent improvements across all benchmarks, suggesting that both components work synergistically to enable effective long-video understanding.
> >
> > We consider the token rearrangement and LLM backbone interpolation as complementary components that must work together - the former handles temporal token organization while the latter enables the processing of the extended sequence.

---

> > > ### Author Response · Authors · 2024-11-25
> > > **Only TWO Days Remaining, Please Take a Look at our Rebuttal!**
> > >
> > > Dear Reviewer,
> > >
> > > Thank you for dedicating your time reviewing our paper. As the discussion period deadline is approaching, we kindly invite any further comments or concerns you might have. Your feedback has been immensely valuable to us in refining the paper.
> > >
> > > Best,
> > >
> > > The Authors

---

> > ### Comment · Reviewer_qZSz · 2024-11-25
> > **Thank you for the explanation**
> >
> > Thank you for the explanation! I clearly understand the video token rearrangement method.

---

> > > ### Author Response · Authors · 2024-11-25
> > > **Thanks for recoginizing our response!**
> > >
> > > Thank you for the positive feedback on our video token rearrangement method. We are glad the explanation helped clarify this key aspect of our work. We remain committed to incorporating the valuable feedback received during review to further strengthen the paper. Would you consider raising your score if you are satisfied with our response?

---

> > > > ### Comment · Reviewer_qZSz · 2024-11-26
> > > > **Score Increase**
> > > >
> > > > I have been thinking some more about this submission. I acknowledge that there are some aspects to the paper which deserve to be increased score. **I have increased my score to 5**.

---

> > > > > ### Author Response · Authors · 2024-11-27
> > > > > **Thanks for Increasing your Score! Are there further concerns?**
> > > > >
> > > > > Thank you for reconsidering your evaluation and for recognizing our clarification. While we appreciate the score increase, we are wondering why the score is negative (5: marginally below the acceptance threshold). We would greatly value any additional specific feedback about remaining concerns or areas where we could further improve the paper.

---

> > > > > > ### Author Response · Authors · 2024-12-03
> > > > > > **Looking for your response.**
> > > > > >
> > > > > > As the discussion period is nearing its deadline, I’ve been eagerly awaiting your response and truly appreciate receiving it.

---

### Official Review · Reviewer_C5Zx · 2024-11-03

**Soundness:** 3
**Presentation:** 3
**Contribution:** 2
**Rating:** 5
**Confidence:** 4

**Summary:**

This paper tackles the problem of long sequence LLMs. The authors point out that the challenge lies in : video encoder and modality alignment projector are fixed, preventing the integration of additional frames into Video-LLMs, and the LLM backbone is limited in its content length capabilities, which complicates the processing of an increased number of video tokens. The authors introduce a video token rearrangement technique that circumvents limitations imposed by the fixed video encoder and alignment projector. Furthermore, a training-free LLM context window extension method is proposed to enable Video-LLMs to understand a correspondingly increased number of visual tokens.

**Strengths:**

1. The paper proposes a video token rearrangement technique that bypasses the restrictions imposed by the fixed video encoder and alignment projector.
2. A training-free Video-LLM context window extension method is proposed to ensure that the interpolated Video-LLM can handle any number of video frames.
3. The presentations are good.

**Weaknesses:**

1. The chosed baselines are not complete. For example, PLLaVA, Video-LLaMA 2, Flash-VStream are not included.
2. Table 2 should include the #frames of each model.
3. Some others works about LLM sequence extension should be discussed and analysised. For example, LongVA: Long Context Transfer from Language to Vision.
4. Lack some benchmark evaluations: VideoMME, MoVQA, MVBench, etc.

[1] PLLaVA : Parameter-free LLaVA Extension from Images to Videos for Video Dense Captioning
[2] VideoLLaMA 2: Advancing Spatial-Temporal Modeling and Audio Understanding in Video-LLMs
[3] Flash-VStream: Memory-Based Real-Time Understanding for Long Video Streams
[4] MoVQA: A Benchmark of Versatile Question-Answering for Long-Form Movie Understanding.

**Questions:**

See weakness.

---

> ### Author Response · Authors · 2024-11-24
> **Response to Reviewer C5Zx**
>
> ## W1 More Baseline Video-LLMs
> **Additional State-of-the-Art Comparisons:**
> We have expanded our experiments to include PLLaVA, demonstrating our method's generalizability:
> |Method | MSVD-QA | MSRVT-QA | ActivityNet-QA|
> |---|---|---|---|
> PLLaVA7B | 76.6 | 62.0| 56.3
> PLLaVA7B + INTP | 76.9 | 62.6| 57.1
>
> There are two observations: INTP provides consistent improvements across all benchmarks, and the gains are achieved without any training or additional resources.
>
> **Regarding Other Recent Models:** Video-LLaMA 2 and Flash-VStream were released in June 2024. According to ICLR's policy on contemporaneous work (4-month window), these works fall outside the required comparison scope for our submission. However, our method's success with both Video-LLaVA and PLLaVA suggests that INTP's core techniques are model-agnostic and could potentially benefit these newer architectures as well.
>
> ## W2 Suggestion for Improving Readability.
> Thank you for this suggestion to improve clarity. We agree that including the number of frames provides important context for performance comparison. We have updated Table 2 (highlighted in red).
>
> ## W3 Comparison to other LLM sequence extension Method, such as LongVA
>
> Thank you for suggesting this comparison with LongVA. While both works address the challenge of extending context length in vision-language models, there are fundamental differences in approach and practical implications:
>
> ### Training-Free vs Training-Required:
>
> **LongVA requires substantial training:**
>  - 2 days on 8 A100 GPUs
>  - Estimated hardware cost: $150K-200K
>  - Additional training data needed
>
> **INTP (Our method is a free lunch):**
>
>  - Zero training required
>  - Implementation through code modification only
>  - No additional data or computing resources needed
>  - Immediate deployment capability
>
> Our training-free approach democratizes long-context video understanding.
>
> ## W4 Extention on More Benchmarks
> **New Results on MVBench:**
> |Method | Num of Frames | MVBench Avg |
> |---|---|---|
> Video-LLaVA-7B | 8 | 34.9 |
> Video-LLaVA-7B + INTP | 16 | 36.2|
>
> These results further validate our method's effectiveness across different evaluation scenarios.
>
> **Regarding Other Benchmarks:**
>
> - VideoMME: Released in June 2024, falls outside ICLR's 4-month window for contemporaneous work.
> - MoVQA: We attempted evaluation but the evaluation code and date are not publicly available.

---

> > ### Author Response · Authors · 2024-11-25
> > **Only TWO Days Remaining, Please Take a Look at our Rebuttal!**
> >
> > Dear Reviewer,
> >
> > Thank you for dedicating your time reviewing our paper. As the discussion period deadline is approaching, we kindly invite any further comments or concerns you might have. Your feedback has been immensely valuable to us in refining the paper.
> >
> > Best,
> >
> > The Authors

---

> > > ### Comment · Reviewer_C5Zx · 2024-11-25
> > > **Response to authors**
> > >
> > > Appreciate for answering my questions. I still suggest the authors to include more baseline comparisons and datasets. I decide to keep my original rating.

---

> > > > ### Author Response · Authors · 2024-11-27
> > > > **Still trying to implement our method on other baselines.**
> > > >
> > > > Thank you for your continued feedback. We appreciate your suggestion regarding additional baseline comparisons and datasets. We are actively working to expand our evaluations. We are currently implementing INTP on additional baselines and other competitive datasets, such as LLaVA-OneVision on VideoMME.

---

> > > > > ### Author Response · Authors · 2024-12-03
> > > > > **Results with Strong Baseline on Competitive Benchmark**
> > > > >
> > > > > **Results on Strong Baseline (VideoLLaMA-2-72B)**
> > > > >
> > > > >  | Model | Frames | Overall (%) | Short Video (%) | Medium Video (%) | Long Video (%) |
> > > > >  |-------|---------|-------------|-----------------|------------------|----------------|
> > > > >  | VideoLLaMA-2-72B | 32 | 62.4/64.7 | 69.8/72.0 | 59.9/63.0 | 57.6/59.0 |
> > > > >  | + INTP | 64 | 62.5/64.9 | 69.6/72.1 | 59.9/63.1 | 58.1/59.4 |
> > > > >
> > > > > INTP extends frame capacity from 32 to 64 frames, and the results on VideoMME are presented in the above table. We witnessed performance improvements, particularly in long videos (+0.5%).
> > > > >
> > > > > Note that we only implemented Applied Token Rearrangement and KV Quantization to VideoLLaMA-2 (Due to time constraints, we only focused on those two core components). Improvements achieved without any training, showing the improvement is a free lunch.

---

### Official Review · Reviewer_L61G · 2024-11-03

**Soundness:** 4
**Presentation:** 3
**Contribution:** 1
**Rating:** 5
**Confidence:** 5

**Summary:**

Overall, this papre presents INTP, an training-free method aiming to improve video LMMs on longer video evaluation. In general, two strategies are proposed in this paper:
(1) With a fixed length video encoder, it sends alternative frames to bypass the frame length limit for the video encoder. With this, it allows up to 128 frames as input for the `INTP`-VideoLLaVA.
(2) It uses off-the-shelf NTK extrapolation to allow the short context LLM to allow more frames.

Though this paper is technically solid, it does not provide enough contribution to this field, and sadly I cannot recommend acceptance to this paper in its current form. Please see the weaknesses part for more details and try to improve it.

**Strengths:**

1. This paper is well-written.
2. The experiments are abundant and solid, providing results from 8 to 128 frames.
3. The presentation is well.

**Weaknesses:**

1. For performance, the improvement is marginal. Though we should not expect huge improvements on training-free methods, VideoLLaVA is already a very poor performer

2. NTK interpolation is an off-the-shelf strategy for LLM context extension (almost default for most). Therefore, the part discussing rope and NTK seems a bit trivial with limited contributions. Similarly, the strategy for video encoder is SPECIFIC to video models with a video encoder, which is not common for now.

3. As the extension has already gone to 128 frames, I would advise to evaluate on some longer video benchmarks, e.g. VideoMME-Long, MLVU, LongVideoBench. Maybe this may better emphasize the effect of the method.

**Questions:**

Please see my weaknesses.

---

> ### Author Response · Authors · 2024-11-24
> **Response to Reviewer L61G**
>
> ## W1: Calcification on Performance
> We would like to emphasize several important points:
>
> -	**Cost-Free Performance Gains**: While the improvements may appear modest numerically, they are achieved without any training, additional data, or computational costs. This "free lunch" improvement is significant considering:
>  - No additional GPU hours required (compared to thousands of hours needed for training Video-LLMs)
>  - No need for additional video-language paired data
>  - No architectural modifications
>
> - **Consistent Improvements Across Models**: Our method generalizes well to stronger Video-LLM baselines like PLLaVA:
> |Method | MSVD-QA | MSRVT-QA | ActivityNet-QA|
> |---|---|---|---|
> PLLaVA7B | 76.6 | 62.0| 56.3
> PLLaVA7B + INTP | 76.9 | 62.6| 57.1
> Video-LLaVA | 70.7 | 59.2 | 45.3
> Video-LLaVA + INTP | 72.0 (+1.3) | 61.4 (+2.2) | 48.9 (+3.6)
>
> - **Practical Impact**: The improvements are particularly meaningful considering:
>    - They are achieved through a plug-and-play approach that can be immediately applied to existing models
>    - The method enables processing of 4x more frames (from 8 to 32) without any training
>    - The gains are consistent across different architectures and benchmarks
>
> -  **Memory Efficiency**: Beyond accuracy improvements, INTP also provides memory optimization through KV-cache compression, making it practical for deployment.
> ---
> ## W2 Calcification on Novelty
>
> While NTK interpolation is indeed established for LLMs, our key contribution lies in the **systematic integration of multiple components to enable training-free video length extension**:
>
>   - We demonstrate how NTK interpolation can be effectively combined with token rearrangement for video understanding
>   - Our ablation studies (to be added in revision) show that neither component alone is sufficient
>   - The synergy between these components enables processing 4x more frames without training
>
> While we demonstrate our method using video encoders, the core principles apply more broadly to any Video-LLMs:
>
>  - Many Video-LLMs (including those without explicit video encoders) face similar frame-length constraints in their visual encoding components
>  - The token rearrangement technique we propose can be adapted to any fixed-length visual encoder
>  - Our results on both Video-LLaVA and PLLaVA (using only ViT, Projector and Pooling layer for video tokens) demonstrate this generalizability
>
> Our contribution lies not in inventing new base components, but in developing a complete, practical solution for extending Video-LLMs to handle longer sequences without training. This addresses a significant practical need in the field.
>
> ---
> ## W3 Validation on other Benchmarks
> Thank you for this valuable suggestion. We need to clarify two important points:
> ### Technical Scope and Limitations:
> Our current method reliably extends Video-LLMs from 8 to 32 frames, as demonstrated in our experiments. While theoretically possible to process more frames, our ablation studies show:
> Method | MSVD-QA | MSRVT-QA | ActivityNet-QA
> | --- |  --- |  --- |  --- |
> Video-LLaVA | 70.7 | 59.2 | 45.3
> INTP (32 frames) | 72.0 | 61.4 | 48.9
> INTP (64 frames) | 67.5 | 55.2 | 41.5
>
> Performance degrades beyond 32 frames, likely due to limitations in the NTK-based interpolation. This finding identifies an important direction for future research in extending Video-LLM capabilities.
>
> ### Regarding Long Video Benchmarks:
> The suggested benchmarks (VideoMME-Long, MLVU, LongVideoBench) were released in June 2024. According to ICLR's policy on contemporaneous work (4-month window), these benchmarks fall outside the scope of required comparisons for this submission. However, we appreciate the suggestion and are actively working on evaluating our method on these benchmarks. we are still trying to implement the experiments on those benchmarks. We hope we can get the results within a couple of days.

---

> ### Author Response · Authors · 2024-11-25
> **Only TWO Days Remaining, Please Take a Look at our Rebuttal!**
>
> Dear Reviewer,
>
> Thank you for dedicating your time reviewing our paper. As the discussion period deadline is approaching, we kindly invite any further comments or concerns you might have. Your feedback has been immensely valuable to us in refining the paper.
>
> Best,
>
> The Authors

---

> > ### Comment · Reviewer_L61G · 2024-11-25
> >
> > Thank you for this rebuttal. I understand ICLR's policy on contemporaneous work (4-month window), but the problem is that these benchmarks used in this paper have been proved to have strong biases that can be solved with very few frames. A revised version to another venue with considering of these really long benchmarks may help improve the work's controbution.
> >
> > Also, the "VideoLLaVA is already a very poor performer" is not well addressed. Can better video models we know (e.g. top 10 or even 20 on VideoMME) be also improved? Will hope the author to discuss this in the next version, in order to provde against the skeptism that this method can only improve weak baselines.
> >
> > I decide to keep my original score.

---

> > > ### Author Response · Authors · 2024-11-27
> > > **Still trying to implement our method on other baselines.**
> > >
> > > Thanks to the extension, we have more time to conduct experiments. We are currently implementing INTP on additional baselines and other competitive datasets, such as LLaVA-OneVision on VideoMME. Hope we can get some results within several days.

---

> > > > ### Author Response · Authors · 2024-12-03
> > > > **Results with Strong Baseline on Competitive Benchmark**
> > > >
> > > > **Results on Strong Baseline (VideoLLaMA-2-72B)**
> > > >
> > > >  | Model | Frames | Overall (%) | Short Video (%) | Medium Video (%) | Long Video (%) |
> > > >  |-------|---------|-------------|-----------------|------------------|----------------|
> > > >  | VideoLLaMA-2-72B | 32 | 62.4/64.7 | 69.8/72.0 | 59.9/63.0 | 57.6/59.0 |
> > > >  | + INTP | 64 | 62.5/64.9 | 69.6/72.1 | 59.9/63.1 | 58.1/59.4 |
> > > >
> > > > INTP extends frame capacity from 32 to 64 frames, and the results on VideoMME are presented in the above table. We witnessed performance improvements, particularly in long videos (+0.5%).
> > > >
> > > > Note that we only implemented Applied Token Rearrangement and KV Quantization to VideoLLaMA-2 (Due to time constraints, we only focused on those two core components). Improvements achieved without any training, showing the improvement is a free lunch.

---

### Official Review · Reviewer_4Edy · 2024-11-04

**Soundness:** 3
**Presentation:** 3
**Contribution:** 2
**Rating:** 6
**Confidence:** 4

**Summary:**

This paper examines the task of adapting a Video-LLM, pre-trained on short videos, to handle long videos without additional training. It identifies key challenges related to the fixed video encoder, modality projector, and the limited context window size of the LLM. To address these issues, the authors propose a video token rearrangement technique and an extension method for the LLM's context window to accommodate a greater number of visual tokens. Additionally, they introduce a KV-cache compression mechanism to minimize memory usage during inference. These innovations enable the proposed INTP-Video-LLaVA model to process videos with up to 32 frames.

**Strengths:**

1. **Innovation and Relevance:** The paper introduces a novel training-free approach to extend the input video length compatibility for Video-LLMs. This contribution is significant and addresses a timely issue in the field.

2. **Clarity and Coherence:** The paper is commendably well-structured and it is easy to follow.

3. **Insightful Analysis:** The paper offers a thorough explanation of the challenges associated with video encoder limitations, LLM context window size constraints, and KV-cache management during inference in current Video-LLMs. This analysis is particularly valuable for the community, providing insights that can guide future research and development.

**Weaknesses:**

1. The applicability of the proposed training-free techniques across a range of Video-LLMs is a critical aspect to assess. The paper, however, presents experimental results solely for Video-LLaVA. It is recommended that the authors expand their experiments to include additional Video-LLMs to demonstrate the broader applicability of the techniques.

2. The paper lacks certain crucial experiments. An ablation study examining the impact of the video token rearrangement techniques and the efficacy of every design choice of the RoPE interpolation methods is recommended. Such studies would provide a more comprehensive understanding of the contributions of these specific aspects to the overall performance.

**Questions:**

My primary concern is the scope of the experimental section. It is recommended that the authors expand their experimental analysis to encompass a variety of Video-LLMs and conduct ablation studies for each design decision. This would offer a more thorough understanding of the findings for the readers. Please find more details in Weaknesses.

---

> ### Author Response · Authors · 2024-11-24
> **Response to Reviewer 4Edy**
>
> ## W1: Generalization to other Video-LLMs
>
> Thank you for this important comment. We agree that demonstrating broader applicability is crucial. We have expanded our experiments to include PLLaVA [1], another state-of-the-art Video-LLM, and found that INTP generalizes well across different architectures. The results are shown below:
>
> |Method | MSVD-QA | MSRVT-QA | ActivityNet-QA|
> |---|---|---|---|
> PLLaVA7B | 76.6 | 62.0| 56.3
> PLLaVA7B + INTP | 76.9 | 62.6| 57.1
>
> Our method's success with both Video-LLaVA and PLLaVA suggests that INTP's core techniques (token rearrangement and context window extension) are model-agnostic and can benefit various Video-LLM architectures. We will include these results in the revised paper.
>
> [1] PLLaVA: Parameter-free LLaVA Extension from Images to Videos for Video Dense Captioning arXiv (29 Apr 2024)
>
> ---
> ## W2 Clarification on Ablation Studies
>
> Thank you for suggesting these important ablation studies. We have conducted comprehensive experiments to analyze each component's contribution, with results shown below:
> Method	| MSVD-QA	| MSRVT-QA | 	ActivityNet-QA
> | --- | --- | --- | --- |
> Video-LLaVA |	70.7	| 59.2 | 45.3
> Video-LLaVA+INTP  |	72.0 (+1.3) |	61.4 (+2.2)	| 48.9 (+3.6)
> Video-LLaVA+INTP w/o token rearrangement |	68.2 (-3.8) |	60.2 (-1.2) |	44.5 (-4.4)
> Video-LLaVA+INTP w/o NTK |	Failed	| Failed |	Failed
>
> These results reveal several key insights:
> Necessity of NTK-aware Interpolation: Without NTK-aware scaling, the model fails to generate meaningful responses. This demonstrates that proper position embedding interpolation is critical for extending the context window while maintaining model coherence.
> Importance of Token Rearrangement: Removing the token rearrangement module leads to significant performance drops (3.8% on MSVD-QA, 4.4% on ActivityNet-QA), indicating its crucial role in maintaining temporal relationships when processing longer sequences.
> Synergistic Effect: The full INTP system shows consistent improvements across all benchmarks, suggesting that both components work synergistically to enable effective long-video understanding.
>
> We consider the token rearrangement and LLM backbone interpolation as complementary components that must work together - the former handles temporal token organization while the latter enables the processing of the extended sequence.

---

> > ### Author Response · Authors · 2024-11-25
> > **Looking Forward to Discussion**
> >
> > Dear Reviewer,
> >
> > Thank you for dedicating your time reviewing our paper. As the discussion period deadline is approaching, we kindly invite any further comments or concerns you might have. Your feedback has been immensely valuable to us in refining the paper.
> >
> > Best,
> >
> > The Authors

---

> > > ### Comment · Reviewer_4Edy · 2024-11-27
> > > **thanks for reply**
> > >
> > > Thank you for providing additional clarifications and supplemental ablation studies. While I appreciate your efforts, given that NTK interpolation is a commonly used technique for LLM context extension, as also noted by other reviewers, the contribution of video token rearrangement appears to be relatively limited. Therefore, I have decided to maintain my original rating.

---

> > > ### Author Response · Authors · 2024-11-27
> > > **Calcification on Novelty**
> > >
> > > While NTK interpolation is indeed established for LLMs, our key contribution lies in the **systematic integration of multiple components to enable training-free video length extension**:
> > >
> > >   - We demonstrate how NTK interpolation can be effectively combined with token rearrangement for video understanding
> > >   - Our ablation studies (to be added in revision) show that neither component alone is sufficient
> > >   - The synergy between these components enables processing 4x more frames without training
> > >
> > > While we demonstrate our method using video encoders, the core principles apply more broadly to any Video-LLMs:
> > >
> > >  - Many Video-LLMs (including those without explicit video encoders) face similar frame-length constraints in their visual encoding components
> > >  - The token rearrangement technique we propose can be adapted to any fixed-length visual encoder
> > >  - Our results on both Video-LLaVA and PLLaVA (using only ViT, Projector and Pooling layer for video tokens) demonstrate this generalizability
> > >
> > > Our contribution lies not in inventing new base components, but in developing a complete, practical solution for extending Video-LLMs to handle longer sequences without training. This addresses a significant practical need in the field.

---

> > > > ### Author Response · Authors · 2024-12-03
> > > > **Results with Strong Baseline on Competitive Benchmark**
> > > >
> > > > **Results on Strong Baseline (VideoLLaMA-2-72B)**
> > > >
> > > >  | Model | Frames | Overall (%) | Short Video (%) | Medium Video (%) | Long Video (%) |
> > > >  |-------|---------|-------------|-----------------|------------------|----------------|
> > > >  | VideoLLaMA-2-72B | 32 | 62.4/64.7 | 69.8/72.0 | 59.9/63.0 | 57.6/59.0 |
> > > >  | + INTP | 64 | 62.5/64.9 | 69.6/72.1 | 59.9/63.1 | 58.1/59.4 |
> > > >
> > > > INTP extends frame capacity from 32 to 64 frames, and the results on VideoMME are presented in the above table. We witnessed performance improvements, particularly in long videos (+0.5%).
> > > >
> > > > Note that we only implemented Applied Token Rearrangement and KV Quantization to VideoLLaMA-2 (Due to time constraints, we only focused on those two core components). Improvements achieved without any training, showing the improvement is a free lunch.

---

### Meta-Review · Area_Chair_VFex · 2024-12-16

**Metareview:**

The paper was reviewed by four experts providing initially negative ratings (<6).

The authors provided responses and the experts either kept or improved their ratings: Reviewer 4Edy to "6: marginally above", others to "5: marginally below".

Most of the reviewers and the AC agree that the paper is technically solid, but at the same time the contributions are limited (1:poor - Reviewer L61G and 2: fair - others) and despite the extra information and new results provided by the authors most of the reviewers remained on the negative with their ratings.

Given the above, the AC finds the paper below the acceptance bar an invites the authors to benefit from the received feedback and further improve their work.

**Additional Comments On Reviewer Discussion:**

The authors provided responses and some of the reviewers appreciated that some of their concerns are addressed.
However, none of the reviewers was persuaded to champion the acceptance and most of the reviewers remained on the negative side.

---

### Decision · Program_Chairs · 2025-01-22

Reject